# Safety outcomes when switching between biosimilars and reference biologics: A systematic review and meta-analysis

Thomas M. Herndon[1]*, Cristina Ausin[1], Nina N. Brahme[1], Sarah J. Schrieber[1], Michelle Luo[1], Frances C. Andrada[1], Carol Kim[1], Wanjie Sun[2], Lingjie Zhou[2], Stella Grosser[2], Sarah Yim[1], M. Stacey Ricci[1]

1 Office of Therapeutic Biologics and Biosimilars, Office of New Drugs, Center for Drug Evaluation and Research, U.S. Food and Drug Administration, Silver Spring, Maryland, United States of America, 2 Division of Biometrics VIII, Office of Biostatistics, Office of New Drugs, Center for Drug Evaluation and Research, U.S. Food and Drug Administration, Silver Spring, Maryland, United States of America

* thomas.herndon@fda.hhs.gov

**Data Availability Statement:** All relevant data are within the manuscript and its Supporting Information files.

## Abstract

Biosimilars are increasingly available for the treatment of many serious disorders, however some concerns persist about switching a patient to a biosimilar whose condition is stable while on the reference biologic. Randomized controlled studies and extension studies with a switch treatment period (STP) to or from a biosimilar and its reference biologic were identified from publicly available information maintained by the U.S. Food and Drug Administration (FDA). These findings were augmented with data from peer reviewed publications containing information not captured in FDA reviews. Forty-four STPs were identified from 31 unique studies for 21 different biosimilars. Data were extracted and synthesized following PRISMA guidelines. Meta-analysis was conducted to estimate the overall risk difference across studies. A total of 5,252 patients who were switched to or from a biosimilar and its reference biologic were identified. Safety data including deaths, serious adverse events, and treatment discontinuation showed an overall risk difference (95% CI) of -0.00 (-0.00, 0.00), 0.00 (-0.01, 0.01), -0.00 (-0.01, 0.00) across STPs, respectively. Immunogenicity data showed similar incidence of anti-drug antibodies and neutralizing antibodies in patients within a STP who were switched to or from a biosimilar to its reference biologic and patients who were not switched. Immune related adverse events such as anaphylaxis, hypersensitivity reactions, and injections site reactions were similar in switched and non-switched patients. This first systematic review using statistical methods to address the risk of switching patients between reference biologics and biosimilars finds no difference in the safety profiles or immunogenicity rates in patients who were switched and those who remained on a reference biologic or a biosimilar.

**Funding:** The author(s) received no specific funding for this work.

**Competing interests:** The authors have declared that no competing interests exist.

## Introduction

Biological products play a central role in medicine, offering safe and effective treatments for many serious disorders [1]. Biosimilar products (biosimilars, BP) are biological products that are highly similar and without clinically meaningful differences to a corresponding FDA-approved reference product[2]. Increasing the availability and use of biosimilars is an important public health strategy for reducing drug costs and increasing the availability of biological products to underserved populations [3]. Historically, effective price competition was lacking for biological products in the U.S. due to the absence of a regulatory pathway specific to the approval of biosimilars. In 2010 with the passage of the Biologics Price Competition and Innovation Act (BPCI Act), the U.S. Food and Drug Administration (FDA) was granted authority to implement an abbreviated approval pathway for biosimilars. Since the enactment of the BPCI Act, FDA has approved forty biosimilars corresponding to eleven different reference products [4]. Many approved biosimilars have yet to be marketed in the U.S. due to patent restrictions, but savings on medication costs correlate with biosimilar usage [5]. A significant increase in biosimilar availability is expected in 2023 as biosimilars to adalimumab begin to be marketed.

Despite the adoption of biosimilars in many therapeutic areas, concerns persist regarding switching a patient to a biosimilar whose condition is stable while on the reference biologic [6]. Among healthcare stakeholders, these concerns are more prevalent among prescribers and, as a results, their patients. Safety and efficacy issues associated with switching between a reference biologic and a biosimilar have been addressed in several reviews of publicly available studies [7–10] and a recent publication from the European Medicines Agency (EMA) utilizing data from European Public Assessment Reports and postmarketing safety surveillance reports from the EMA for approved monoclonal antibodies [10]. While providing comprehensive characterizations of safety profiles after switching between biosimilars and their corresponding reference biologics, these reviews are primarily descriptive without the use of formal statistical methods in the data synthesis. The inclusion of real-world evidence in these reviews may confound interpretation of results due to the recognized nocebo effects observed in real world studies of biosimilars [11].

While the purpose of studies employing a switching period is to explore the potential for safety concerns associated with a single switch between a reference biologic and a candidate biosimilar, confusion persists about the role of "switching studies" in obtaining the "interchangeability" designation [12]. This designation created by the BPCI Act distinguishes an interchangeable biosimilar product as one that may be substituted at a retail pharmacy in a manner akin to generic substitution, whereas a biosimilar must be prescribed by name [13]. FDA has recommended that studies include one or more switches between the reference biologic and the biosimilar to address concerns about increased risk of immunogenicity for patients who chronically use certain reference biologics and are at risk in terms of hypersensitivity, anaphylaxis, neutralizing antibody, or other reactions. Whether such studies are needed to assess risk from one or more switches is a matter of ongoing discussion among regulatory agencies including FDA [10, 14]. To date, a systematic review using statistical methods has not been performed to address the value of studies with multiple switches together with single switches using statistical methods.

Main objectives of a regulatory safety review are to identify and closely examine adverse events that suggest significant concerns with a drug, specifically, adverse reactions severe enough to prevent the use of the drug altogether, to limit its use, or require special risk management efforts [15]. Deaths, non-fatal serious adverse events (SAE), and discontinuations of the study drug due to an adverse event (discontinuations) are the primary events across

development programs severe enough to preclude continued use. We examined the occurrence of safety events following a switch to or from a biosimilar and its reference biologic in all identified controlled clinical studies that included a biosimilar approved by FDA. While safety and immunogenicity were assessed after switching, efficacy typically was not because the primary efficacy determination of a proposed biosimilar occurs earlier in a clinical study prior to the switching period. Publicly available information extracted from FDA reviews of Biologics License Applications (BLAs) was augmented with additional safety data (not included in FDA reviews) from published reports. Data synthesis was performed to summarize results and to quantify single estimates of risk when switching. The results demonstrate that the incidence and nature of safety events observed in patients who were switched to or from a biosimilar product and its corresponding reference biologic once or multiple times are not statistically different from those in patients who were not switched.

## Materials and methods

This systematic review was performed and is reported in accordance with the Preferred Reporting Items for Systematic Reviews and Meta-Analyses (PRISMA) statement [16] (S1 Table in S1 File).

### Searches and source identification

Searches were performed for the period from 01 January 2000 through 31 December 2022; FDA databases containing publicly available information were reviewed for all clinical studies submitted as part of a BLA for an approved biosimilar. Information not captured in FDA reviews was identified by searching the Embase. MEDLINE, and PubMed databases (additional details in S1 File).

### Study screening and inclusion

Two reviewers independently screened titles, abstracts, and full text of articles for eligibility. Disagreements on eligibility were resolved by a third reviewer. The eligibility criteria for the systematic review were: 1.) Study must be a randomized controlled study or an extension study from a randomized controlled study, 2.) The biosimilar in the study must be approved in the United States, 3.) There must be at least one Switch Treatment Period (STP) that included one or more switches of a biosimilar for a corresponding reference biologic or a reference biologic for a biosimilar, 4.) Study must have safety data available, and 5.) If all relevant data from a study was contained in the FDA Review, a published report from the same study was not eligible for inclusion.

### Data extraction and quality assessment

The STP of a study was used as the fundamental unit for data extraction because eligible studies could have one or more STPs containing a switch that were eligible for inclusion. Each comparison within a STP was given a unique identifier (e.g., STP-01). Two reviewers independently extracted the following data from each STP: 1.) Reference biologic, 2.) Biosimilar, 3.) Control and test arms, 4.) Associated BLA number, 5.) Study identifier, 6.) Study design including randomization, masking, population, switching information including number of switches, 7.) Study participant demographics, 8.) Safety data, and 9.) Immunogenicity data (Table 1).

**Table 1. Characteristics of switch treatment periods.**

| Switch Treatment Period | Biosimilar Product Nonproprietary Name | National Clinical Trial Number | Design | Patient Population | Treatment Before STP (Weeks) | STP Length (Weeks) | Switches up to and Including STP (n) | No Switch (Control Arm, n) | Switch (Test Arm, n) | Source |
|---|---|---|---|---|---|---|---|---|---|---|
| **adalimumab** | | | | | | | | | | |
| STP-01 | adalimumab-aacf | NCT02660580 | RDB | PS | 16 | 52 | 1 | R-R, 101 | R-B, 101 | FDA Reviews |
| STP-02[a] | adalimumab-adaz | NCT02016105 | RDB | PS | 17 | 18 | 3 | R-R, 127 | R-B, 63 | FDA Reviews |
| STP-03[a] | adalimumab-adaz | NCT02016105 | RDB | PS | 17 | 18 | 3 | B-B, 126 | B-R, 63 | FDA Reviews |
| STP-04[a] | adalimumab-adaz | NCT02016105 | RDB | PS | 35 | 16 | 4 | R-R, 109 | R-B, 56 | FDA Reviews |
| STP-05[a] | adalimumab-adaz | NCT02016105 | RDB | PS | 35 | 16 | 4 | B-B, 106 | B-R, 52 | FDA Reviews |
| STP-06 | adalimumab-adaz | NCT02744755 | RDB | RA | 24 | 24 | 1 | B-B, 159 | R-B, 166 | Wiland 2020 [39] |
| STP-07 | adalimumab-adbm | NCT02137226 | RDB | RA | 24 | 24 | 1 | R-R, 148 | R-B, 146 | FDA Reviews |
| STP-08 | adalimumab-adbm | NA | RDB | PS | 14 | 18 | 3 | R-R, 120 | R-B, 118 | FDA Reviews |
| STP-09 | adalimumab-afzb | NCT02480153 | RDB | RA | 26 | 26 | 1 | R-R, 135 | R-B, 133 | FDA Reviews |
| STP-10 | adalimumab-afzb | NCT02480153 | OLE | RA | 52 | 26 | 2 | B-B, 127 | R-B, 120 | FDA Reviews, Fleischmann 2021 [28] |
| STP-11 | adalimumab-aqvh | NCT02489227 | RDB | PS | 16 | 8 | 1 | R-R, 130 | R-B, 126 | FDA Reviews |
| STP-12 | adalimumab-aqvh | NCT02489227 | OLE | PS | 24 | 24 | 2 | B-B, 113 | R-B, 126 | FDA Reviews |
| STP-13 | adalimumab-atto | NCT01970488 | RDB | PS | 16 | 32 | 1 | R-R, 79 | R-B, 77 | FDA Reviews, Papp 2017 [33] |
| STP-14 | adalimumab-atto | NCT02114931 | OLE | RA | 22 | 72 | 1 | B-B, 230 | R-B, 237 | Cohen 2019 [23] |
| STP-15 | adalimumab-bwwd | NCT02167139 | RDB | RA | 24 | 26 | 1 | R-R, 127 | R-B, 125 | FDA Reviews, Weinblatt 2018 [38] |
| STP-16 | adalimumab-fkjp | NCT02405780 | OLE | RA | 24 | 30 | 1 | R-R, 213 | R-B, 108 | FDA Reviews |
| STP-17 | adalimumab-fkjp | NCT02405780 | OLE | RA | 24 | 30 | 1 | B-B, 216 | B-R, 108 | FDA Reviews |
| STP-18 | adalimumab-fkjp | NCT02405780 | OLE | RA | 54 | 49 | 2 | B-B, 93 | R-B, 190 | FDA Reviews |
| STP-19 | adalimumab-fkjp | NCT02405780 | OLE | RA | 54 | 49 | 2 | B-B, 189 | R-B, 100 | FDA Reviews |
| **epoetin alfa** | | | | | | | | | | |
| STP-20 | epoetin alfa-epbx | NCT01473407 | RDB | CKD | 4 | 24 | 1 | R-R, 304 | R-B, 301 | FDA Reviews, Fishbane 2018 [27] |
| STP-21 | epoetin alfa-epbx | NCT02504294 | ROL | CKD | $\geq$ 16 | 24 | 1 | R-R, 206 | R-B, 212 | Thadhani 2018 [36] |
| **etanercept** | | | | | | | | | | |
| STP-22 | etanercept-szzs | NCT01891864 | RDB | PS | 12 | 18 | 3 | R-R, 151 | R-B, 96 | FDA Reviews |
| STP-23 | etanercept-szzs | NCT01891864 | RDB | PS | 12 | 18 | 3 | B-B, 150 | B-R, 100 | FDA Reviews |
| STP-24 | etanercept-szzs | NCT02638259 | RDB | RA | 24 | 24 | 1 | B-B, 175 | R-B, 166 | Jaworski 2019 [30] |
| STP-25 | etanercept-ykro | NCT01895309 | OLE | RA | 52 | 48 | 1 | B-B, 126 | R-B, 119 | FDA Reviews, Emery 2017 [26] |
| **filgrastim** | | | | | | | | | | |
| STP-26 | filgrastim-sndz | NCT01519700 | RDB | BC | 3 | 15 | 5 | R-R, 52 | R-B and B-R, 109 | Blackwell 2018 [22] |
| **infliximab** | | | | | | | | | | |
| STP-27 | infliximab-adba | NCT01936181 | RDB | RA | 54 | 24 | 1 | R-R, 101 | R-B, 94 | FDA Reviews |
| STP-28 | infliximab-axxq | NCT02937701 | RDB | RA | 22 | 24 | 1 | R-R, 121 | R-B, 119 | FDA Reviews |

(*Continued*)

**Table 1.** (Continued)

| Switch Treatment Period | Biosimilar Product Nonproprietary Name | National Clinical Trial Number | Design | Patient Population | Treatment Before STP (Weeks) | STP Length (Weeks) | Switches up to and Including STP (n) | No Switch (Control Arm, n) | Switch (Test Arm, n) | Source |
|---|---|---|---|---|---|---|---|---|---|---|
| STP-29 | infliximab-dyyb | NCT01571219 | Blinded to previous treatment | RA | 54 | 48 | 1 | B-B, 158 | R-B, 144 | FDA Reviews, Yoo 2017 [41] |
| STP-30 | infliximab-dyyb | NCT01571206 | OLE | AS | 54 | 48 | 1 | B-B, 90 | R-B, 84 | FDA Reviews, Park 2017 [34] |
| STP-31 | infliximab-dyyb | NCT02148640 | RDB | CD, UC, SA, RA, PA, PS | ≥ 24 | 52 | 1 | R-R, 241 | R-B, 241 | Jørgensen 2017 [31] |
| STP-32 | infliximab-dyyb | NCT02148640 | RDB | CD, UC, SA, RA, PA, PS | 52 | 26 | 2 | B-B, 197 | R-B, 183 | Goll 2019 [29] |
| STP-33 | infliximab-dyyb | NCT02096861 | RDB | CD | 30 | 24 | 1 | R-R, 54 | R-B, 55 | Ye 2019 [40] |
| STP-34 | infliximab-dyyb | NCT02096861 | RDB | CD | 30 | 24 | 1 | B-B, 56 | B-R, 55 | Ye 2019 [40] |
| STP-35 | infliximab-qbtx | NCT02222493 | RDB | RA | 30 | 54 | 1 | R-R, 143 | R-B, 143 | FDA Reviews |
| STP-36 | infliximab-qbtx | NCT02222493 | OLE | RA | 54 | 24 | 2 | B-B, 126 | R-B, 126 | Cohen 2020 [25] |
| **insulin glargine** | | | | | | | | | | |
| STP-37 | insulin glargine-yfgn | NCT02666430 | ROL | DM | 52 | 36 | 3 | R-R, 63 | R-B, 64 | FDA Reviews |
| **rituximab** | | | | | | | | | | |
| STP-38[a] | rituximab-abbs | NCT02149121 | ROL | RA | 48[a] | 24 | 1 | R-R, 64 | R-B, 109 | FDA Reviews, Shim 2019 [35] |
| STP-39 | rituximab-abbs | NCT01873443 | ROL | RA | 24 to 48 | 24 | 1 | B-B, 38 | R-B, 20 | FDA Reviews |
| STP-40 | rituximab-abbs | NCT02260804 | RDB | FL | 64 | 52 | 1 | B-B, 110 | R-B, 103 | Kwak 2022 [32] |
| STP-41[a] | rituximab-arrx | NCT02792699 | ROL | RA | 24 | 24 | 1 | R-R, 104 | R-B, 103 | FDA Reviews |
| STP-42[a] | rituximab-pvvr | NCT01643928 | ROL | RA | ≥ 16 | 9 | 1 | R-R, 62 | R-B, 63 | FDA Reviews |
| STP-43[a] | rituximab-pvvr | NCT01643928 | Blinded randomization | RA | ≥ 16 | ≥ 16 | 2 | B-B, 59 | R-B, 57 | FDA Reviews, Cohen 2018 [24] |
| **trastuzumab** | | | | | | | | | | |
| STP-44 | trastuzumab-anns | NCT01901146 | RDB | BC | ≤ 12 | ≤ 33 | 1 | R-R, 171 | R-B, 171 | FDA Reviews, von Minckwitz 2018 [37] |

AS, Ankylosing Spondylitis; B, Biosimilar Product; BC, Breast Cancer; CD, Crohn Disease; CKD, Chronic Kidney Disease; DM, Type 1 Diabetes Mellitus; FL, Follicular Lymphoma; NCT, National Clinical Trial Number; NA, Not Applicable; OLE, Open Label Extension; PA, Psoriatic Arthritis; PS, Plaque Psoriasis; RA, Rheumatoid Arthritis; RDB, Randomized Double Blind; ROL, Randomized Open Label; SA, Spondylarthritis; STP, Switch Treatment Period; UC, Ulcerative Colitis.

[a]Data included in the submission to justify combining EU and US Reference Biologic arms.

Two reviewers independently assessed risk of bias for each STP using a modified version of the Newcastle-Ottawa Scale [17]. STPs that scored 4 or more on the modified Newcastle-Ottawa Scale were considered to have a low risk of bias [18] (details in S1 File).

## Data synthesis and analysis

Data was synthesized qualitatively and quantitatively. Information on reference biologics, biosimilars, study characteristics such as randomization, blinding, comparators, and demographics of the study populations were collated descriptively. The reference biologic was either the

U.S.-licensed reference product or the corresponding European Union (E.U.)-approved version of the U.S. reference product. Other characteristics of study designs and STPs, such as exposures to related reference biologics or biosimilar products used before the STP, and the length of time of the STP were quantified. Safety data (deaths, serious adverse events, and permanent discontinuations from study drug due to an adverse event) were expressed as the percentage of patients with the respective safety event among those enrolled in the corresponding arm within each STP. Risk differences between Switch and No Switch arms in each STP were calculated for deaths, serious adverse events, and discontinuations. Meta-analyses were performed and forest plots generated using the open-source statistical package, R 4.0.3. A test for homogeneity of risk difference across STPs was conducted using a chi-square test [19]. If the risk difference is homogeneous across STPs, then a fixed effect model [20] was used to estimate the overall risk difference and 95% confidence interval (CI). Otherwise, a DerSimonian-Laird random-effects model [21] was used instead. The significance level used is 0.05.

Antidrug antibody data and neutralizing antibody data were presented graphically. For consistency, in data presentation across programs, neutralizing antibody data shown in source materials as a percentage of patients with antidrug antibodies was converted to the percentage of study participants with antidrug antibodies in the treatment arm. Clinical data of adverse events associated with immunogenicity were summarized qualitatively and numerically.

Missing safety data was rare, but in a few cases anonymized data was included from clinical study reports when isolated time points were missing from publicly available reports.

## Results

Screening and selection assessment outcomes for inclusion are depicted in the PRISMA flow diagram (Fig 1). The search process identified 552 records (141 from publicly available FDA reviews, 408 from public databases, and 3 from additional public sources) of which 44 (23 FDA reviews (Table 1, S2 Table in S1 File), 21 publications [22–41]) met eligibility criteria. All included studies were randomized double blind controlled studies or open label extensions of a randomized double blind controlled parent study. All studies included at least one STP during which patients were switched one or more times between a reference biologic and a corresponding biosimilar candidate that was subsequently licensed in the U.S. STPs contained switches from reference biologic to biosimilar or biosimilar to reference biologic provided the STP contained a no switch arm to serve as control. Forty-four distinct STPs were identified from thirty-one unique studies.

### Study characteristics

The included STPs represent twenty-one U.S.-licensed biosimilars corresponding to eight different reference biologics (Table 1, S2 Table in S1 File). The number of unique studies and STPs for each reference biologic with an FDA approved biosimilar are adalimumab (Studies 11, STPs 19), epoetin-alfa (Studies 2, STPs 2), etanercept (Studies 3, STPs 4), filgrastim (Studies 1, STPs 1), infliximab (Studies 7, STPs 10), insulin-glargine (Studies 1, STPs 1), rituximab (Studies 5, STPs 6), and trastuzumab (Studies 1, STPs 1).

Twenty-eight STPs contained a single switch and 16 STPs had multiple switches within the STP (Tables 1 and S2 and S2 Fig in S1 File). The most common switching pattern observed was a single switch of patients from the reference biologic to the biosimilar (R-B) compared to patients who were not switched (R-R) (n = 18). Sixteen STPs evaluated multiple switches between the biosimilar and the reference biologic, with a range of 3 to 5 switches within a STP containing multiple switches. Sixteen STPs were identified where patients were exposed to a biosimilar or reference biologic more than once prior to and including the STP. Patients in

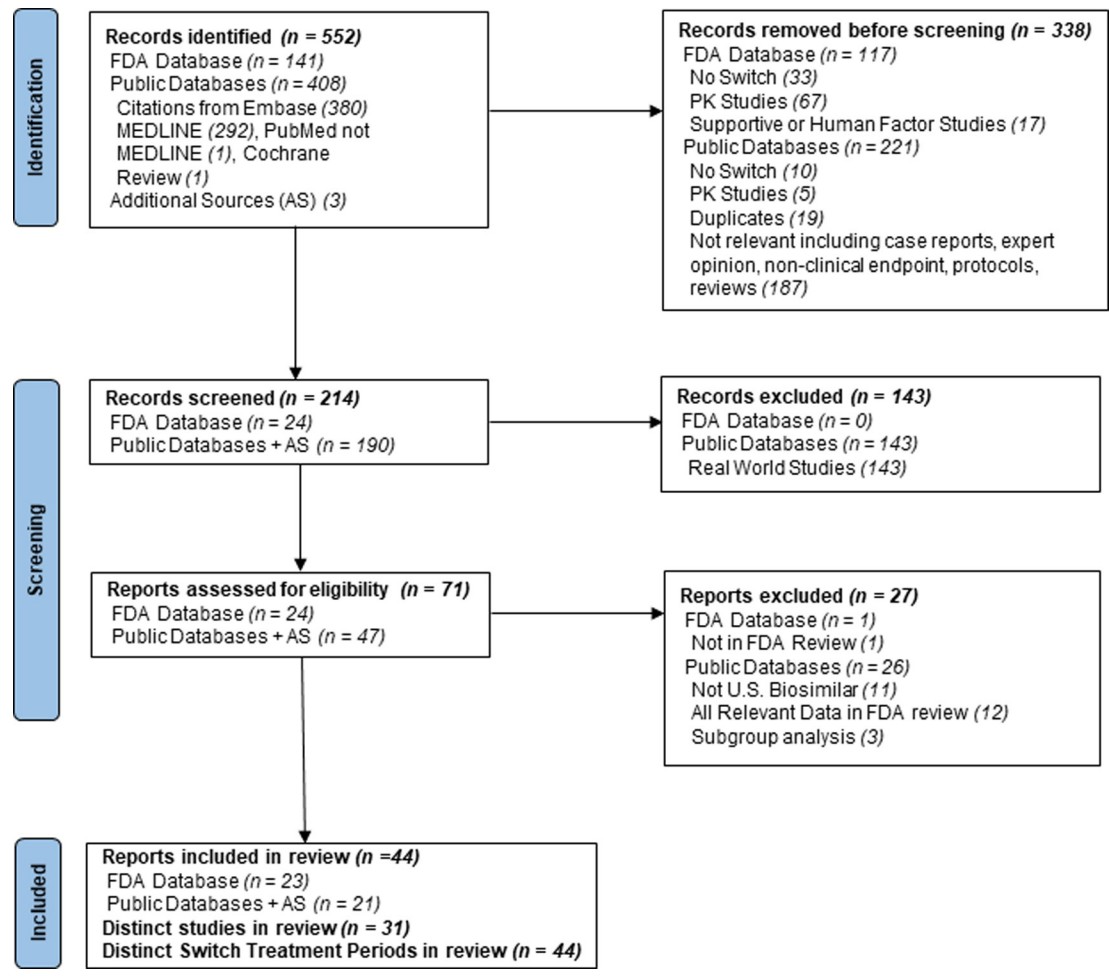

**Fig 1. Evidence screening and selection assessment.**

STPs occurring towards the end of a study or in an extension study were more likely to have been exposed to multiple switches. The length of time on a study drug before a STP ranged from 3 to 64 weeks and the length of time of the STP (including follow-up) ranged from 8 to 72 weeks.

## Patient characteristics

Five thousand two hundred fifty-two patients who underwent at least one switch across all STPs were analyzed (Table 1, S2 Table in S1 File). The patients and diseases are representative of those likely to be treated with a currently approved biosimilar. Study populations are chosen for clinical studies designed to support a biosimilar program based on the likelihood that the population will allow an assessment of clinically meaningful differences following treatment with the proposed biosimilar or the reference product [42]. As a major concern with switching between reference products and their biosimilars is development of an undesired immune response [13], the inclusion of patients with active immune systems in studies intended to support a regulatory action is more common than including patients who are immunosuppressed. For this reason and because the majority of approved biosimilars are TNF inhibitors used to treat patients with inflammatory conditions, patients with rheumatoid arthritis, plaque

psoriasis, Crohn disease, and ulcerative colitis were more frequently enrolled (38 STPs). The remainder of the STPs consisted of patients with chronic kidney disease, breast cancer, type 1 diabetes, and follicular lymphoma.

Baseline demographic characteristics collected include age, sex, race, ethnicity, and BMI were evenly distributed between the reference biologic and biosimilar arms. When reported, the characteristics of the patients at baseline were also evenly distributed between the no switch and switch arms (S3, S4 Tables in S1 File).

## Safety comparisons

Study drug exposure was adequate and equivalent between Switch and No Switch arms across STPs (S5 Table in S1 File). Data on deaths, SAEs, and discontinuations identified as part of this review were compiled (Figs 2–4, S6 Table in S1 File) from source material. A negative point estimate of the risk difference between the Switch and No Switch arms favors No Switch, a positive result favors Switch and a value of zero is associated with no difference.

Meta-analysis across STPs was performed for all three safety outcomes. The risk difference for each outcome was homogeneous across STPs (P-values >0.05, Figs 2–4). For death, there were 21 events out of 5252 subjects in the Switch and 23 out of 5770 in the No Switch arms, with an overall risk difference -0.00 and 95% CI (-0.00, 0.00) (Fig 2). For SAE, 436 subjects had at least one event out of 5252 subjects in the Switch and 433 out of 5770 had at least one SAE in the No Switch arms; overall risk difference -0.00 and 95% CI (-0.01, 0.01) (Fig 3). For treatment discontinuations, 142 subjects had the event out of 5252 in the Switch and 160 out of 5770 in the No Switch arms; overall risk difference -0.00 and 95% CI (-0.01, 0.00) (Fig 4). Figs 2–4 demonstrate that the risk of all three safety events were similar and there was no statistically significant increase in risk of a major safety concern when switching between a reference biologic and a biosimilar.

Additional analyses of the source data were performed to determine if the pooled result was influenced by a design component of the STP. Analyses of any reference product class with five or more STP in the full analysis (adalimumab, infliximab, rituximab) were performed separately: forest plots and summary data for deaths, SAEs, and discontinuations were consistent with the full analysis (S3-S11 Figs in S1 File). Analyses comparing deaths, SAE, and discontinuations when switches were performed from reference biologic to biosimilar (R-B) (S12-S14, S18-S20 Figs in S1 File) or from biosimilar to reference biologic (B-R) (S15-S17 Figs in S1 File) and single switches (S18-S23 Figs in S1 File) or multiple switches (S24-S26 Figs in S1 File) also demonstrated results similar to the full analysis.

## Immunogenicity

Immunological assessments are performed as part of most clinical studies comparing a biosimilar to a reference biologic. For the studies in this review, immunogenicity antibody samples were collected prior to the STP where one or more switches occurred and at the end of the STP (S27 Fig, S7 Table in S1 File). Due to exposure to reference biologics prior to the STP, large proportions of a study population may have positive antidrug antibodies and neutralizing antibody formation before a switch occurs. Study subjects are usually not stratified by the presence or absence of antidrug antibodies prior to a switch. For these reasons and as the assays for antidrug antibodies are not standardized across programs, it is not scientifically appropriate to compare antidrug antibody levels or their neutralizing ability across biosimilar programs, including those having the same reference product.

Clinical events associated with immunogenicity were rare and were not associated with switching (S8 Table in S1 File). In cases where antidrug antibody data were collected at the

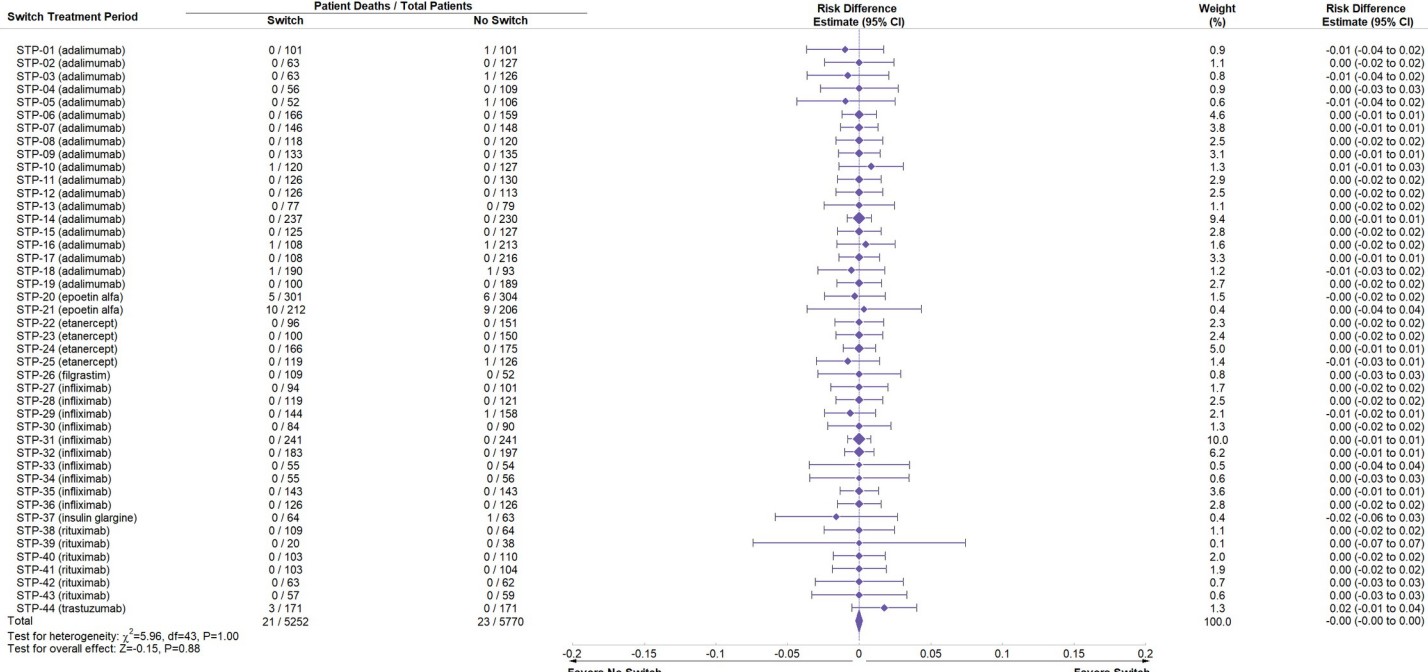

**Fig 2. Meta-analysis of risk difference in death between Switch and No Switch arms following switching across all switch treatment periods.** Meta-analysis was performed of the risk difference for death between Switch and No Switch arms in each switch treatment period (STP). Weight refers to the contribution of each STP to the overall estimate of risk difference, which is based on the inverse of the variance of the respective risk difference. $\chi^2$ and df are used in the chi-square test for homogeneity of risk difference across studies. Z value is used in the normal Z test for whether the overall risk difference is zero. CI is the confidence interval.

time of a clinical event, there was no correlation with the onset of the event and antidrug antibody status (S9 Table in S1 File).

## Bias

STPs were reviewed for bias using a modified version of the Newcastle-Ottawa Scale [17]. STP that scored 4 or more points on the modified Newcastle-Ottawa Scale were considered to have a low risk of bias [18]. All STP were scored 4 or higher indicating low risk of bias (S28 Fig in S1 File).

## Discussion

Based on data from controlled studies or extensions of controlled studies in this systematic review, no differences in terms of major safety parameters such as deaths, SAEs, and discontinuations were observed when patients are switched (to or from a biosimilar and its reference biologic) or not switched (Figs 2–4). The result is the same across all STPs and independent of the reference product class (S3-S11 Figs in S1 File), the direction of the switch (R-B or B-R) (S12-S17 Figs in S1 File), or following one or multiple switches (S18-S26 Figs in S1 File). The incidence of antidrug antibodies and neutralizing antibodies was comparable between No Switch and Switch arms (S27 Fig and S9 Table in S1 File). The study participants in the STPs are representative of patients who would be switched to or from a biosimilar and its reference product and reflect actual use. This result is consistent with empiric evidence and descriptive reviews of switching biosimilars and reference biologics [7, 8, 10].

A successful biosimilar development program begins with a comprehensive and robust comparison of physicochemical and biological functional attributes of the proposed biosimilar

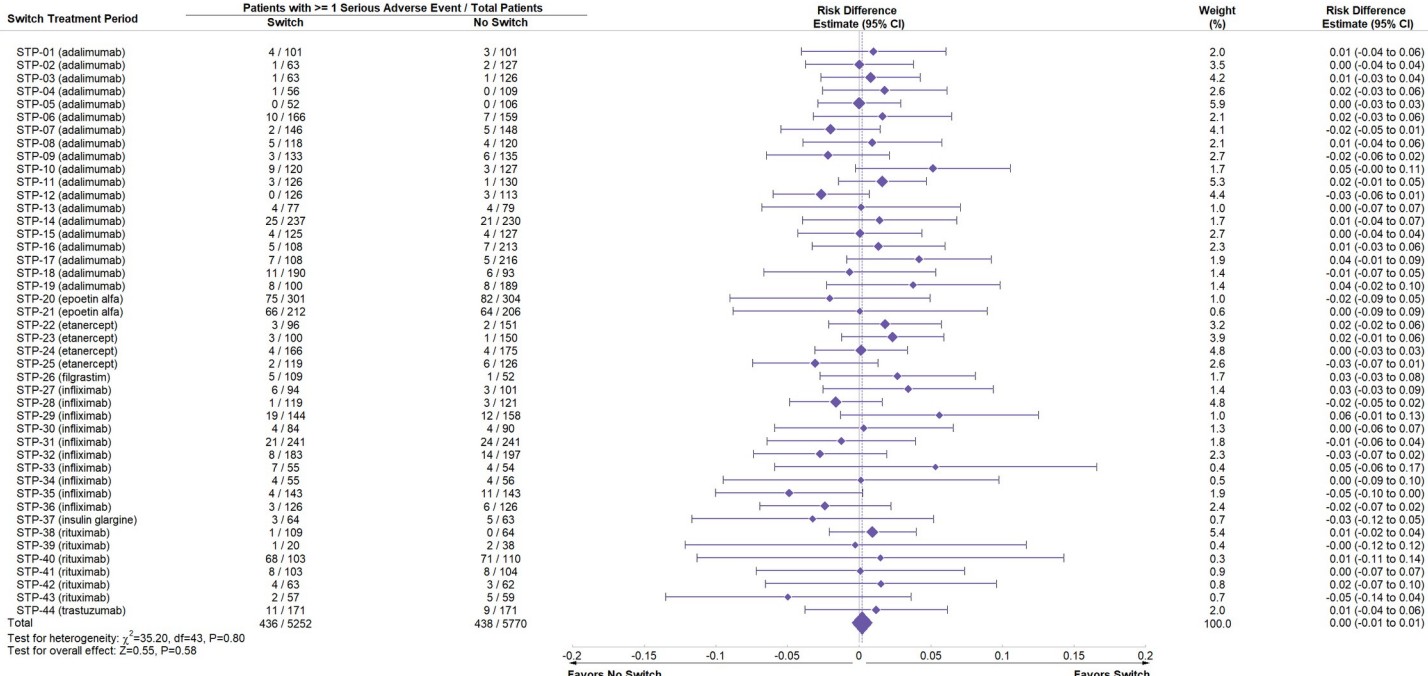

**Fig 3. Meta-analysis of risk difference in serious adverse events between Switch and No Switch arms following switching across all switch treatment periods.** Meta-analysis was performed of the risk difference for one or more serious adverse events between Switch and No Switch arms in each switch treatment period (STP). Weight refers to the contribution of each STP to the overall estimate of risk difference, which is based on the inverse of the variance of the respective risk difference. $\chi^2$ and df are used in the chi-square test for homogeneity of risk difference across studies. Z value is used in the normal Z test for whether the overall risk difference is zero. CI is the confidence interval.

to the reference product. These comparisons make use of state-of-the-art techniques to evaluate key characteristics such as molecular structure, bioactivity, and purity and support the demonstration that a proposed biosimilar is highly similar to its reference product as a condition of licensure. Additional information on the depth of this comparative analytical assessment is provided in the S1 File. As the biosimilar is designed to be analytically highly similar to its reference product, the clinical studies comparing the two products are essentially confirming the expected clinical outcome (i.e., similar PK, safety, PD/efficacy, and immunogenicity). The findings in this report raise several timely questions regarding the data needed to support approval of biosimilars and how to address regulatory requirements unique to the BPCI Act for the "interchangeable" designation.

Biosimilar products with the interchangeable designation, in contrast to biosimilar products without it may be substituted at a retail pharmacy in a manner akin to generic substitution. Biosimilars with and without the designation must be highly similar to and without clinically meaningful differences from the reference product. In addition to being biosimilar to the reference product, the BPCI Act requires that two additional criteria be addressed to be designated interchangeable [12]. The interchangeable product must be expected to produce the same clinical result as the reference product in any given patient, and for a product administered more than once, the risk of switching between a reference product and an interchangeable product must not be greater than the risk of using the reference product without switching. A "switching study" may be performed to address this "switching standard". Such "switching studies" often contain at least two alternating exposures of the proposed interchangeable product and the reference biologic resulting in at least three switches. The

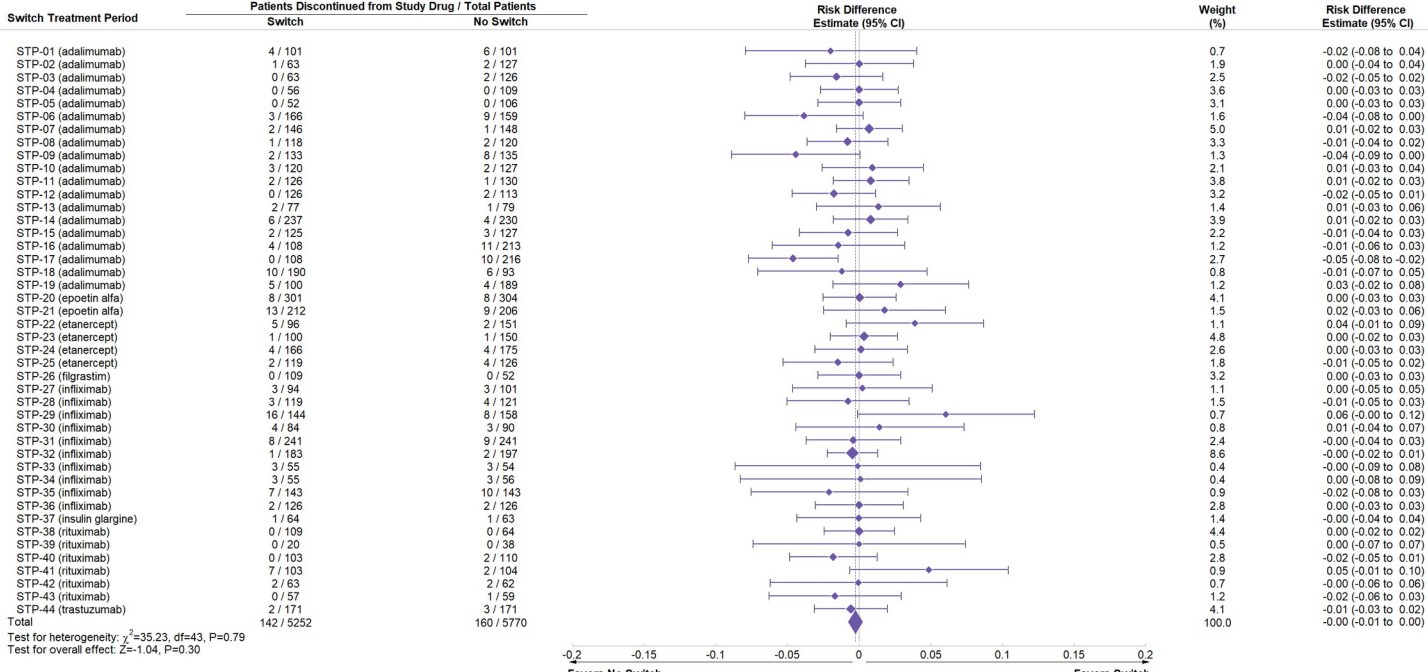

**Fig 4. Meta-analysis of risk difference in discontinuation between Switch and No Switch arms following switching across all switch treatment periods.** Meta-analysis was performed of the risk difference for permanent discontinuation of study drug due to an adverse event between Switch and No Switch arms in each switch treatment period (STP). Weight refers to the contribution of each STP to the overall estimate of risk difference, which is based on the inverse of the variance of the respective risk difference. $\chi^2$ and df are used in the chi-square test for homogeneity of risk difference across studies. Z value is used in the normal Z test for whether the overall risk difference is zero. CI is the confidence interval.

recommended endpoints for a "switching study" are a statistically powered PK comparison and a descriptive comparison of safety [13].

The findings reported here support reducing the regulatory burden of switching studies as the default approach for addressing the switching standard for the interchangeable designation. This work also supports reassessing the need for switches included in clinical studies for candidate biosimilars as an approved biosimilar will be analytically highly similar to its reference product. As familiarity with and understanding of the rigor of the analytical comparisons used to support biosimilar approvals increases, the amount and types of clinical data routinely performed as part of biosimilar development may be reduced, which in turn would reduce the time and cost of development.

Limitations of this review include the small number of patients in the safety evaluations from some source material (Table 1, n range 20–301). While there are fewer patients in some STPs, these differences in STP sample size have been taken into consideration in our meta-analysis. Randomization into STP arms was not consistently balanced. Among the 44 STPs, most (75%) were balanced and 11 STPs (25%) were imbalanced. In most cases where imbalance occurred, more patients were enrolled in the No Switch arm (Table 1, S2 Table in S1 File). However, given that risk is evaluated as the percentage of patients with the respective safety event among those enrolled in the corresponding arm within each STP, a larger sample size in the No Switch arm of an unbalanced designed clinical study should not impact the risk difference results for safety. There is considerable variation in the characteristics of the STPs and in patient demographics. To address this, additional exploratory Logistic regression were performed using summary data to evaluate the potential impact of study heterogeneity on the

comparison of risk in each safety event between the Switch and No Switch groups. The unadjusted OR of Switch vs. No Switch groups for all three safety events were not significantly different from 1 (OR = 1.003 for death, 1.102 for SAE, 0.974 for discontinuation, each p>0.05) (S10-S12 Tables in S1 File). After adjusting for individual study and patient factors, the adjusted OR had a trend of favoring the Switch group for death (OR = 0.73–0.97, each p>0.05), slightly favoring the No Switch group for SAE (OR = 0.967–1.06, each p>0.05), and mostly favoring the Switch group for Discontinuation (OR = 0.955–1.08, each p>0.05). The exploratory Logistic regression did not change our study conclusion. Additional information on the exploratory Logistic regression is contained in the S1 File.

Endpoints related to PK or the effectiveness of a drug that were collected during the STPs were not included in this systematic review. If efficacy data were collected periodically during STPs, it was not powered for an efficacy determination. The efficacy data used to determine equivalent efficacy of a biosimilar to a reference biologic is contained in publicly available FDA reviews and in the publications for the individual studies that led to approval of the biosimilars. Non-statistical descriptions of efficacy data from portions of studies that contain switches are available in published reviews [7–10]. In patients where PK data was collected and analyzed during a switch treatment period, there were no associated differences in efficacy or safety related to changes in PK for patients in the No Switch and Switch arms.

This work addresses one of the medical community's concerns regarding the safety of switching between reference products and corresponding biosimilar products. Data driven materials such as those contained in this report will facilitate efforts to streamline biosimilar development and achieve the full promise of biosimilars.

## Supporting information

**S1 File.**
(DOCX)

**S1 Checklist. Preferred Reporting Items for Systematic reviews and Meta-Analyses (PRISMA) checklist.** Page numbers refer to the page number in the original submitted materials.
(DOCX)

## Author Contributions

**Conceptualization:** Thomas M. Herndon, Cristina Ausin, Nina N. Brahme, Sarah J. Schrieber, Michelle Luo, Sarah Yim, M. Stacey Ricci.

**Data curation:** Thomas M. Herndon, Cristina Ausin, Nina N. Brahme, Sarah J. Schrieber, Michelle Luo, Frances C. Andrada, Carol Kim, Wanjie Sun, Lingjie Zhou.

**Formal analysis:** Wanjie Sun, Lingjie Zhou.

**Investigation:** Thomas M. Herndon, Cristina Ausin, Nina N. Brahme, Sarah J. Schrieber, Michelle Luo, Frances C. Andrada.

**Methodology:** Thomas M. Herndon, Cristina Ausin, Nina N. Brahme, Sarah J. Schrieber, Michelle Luo.

**Supervision:** Sarah Yim, M. Stacey Ricci.

**Validation:** Thomas M. Herndon, Cristina Ausin, Nina N. Brahme, Sarah J. Schrieber, Michelle Luo, Frances C. Andrada, Carol Kim.

**Visualization:** Thomas M. Herndon, Cristina Ausin, Nina N. Brahme, Sarah J. Schrieber, Michelle Luo, Frances C. Andrada, Wanjie Sun, Lingjie Zhou.

**Writing – original draft:** Thomas M. Herndon, Sarah J. Schrieber, M. Stacey Ricci.

**Writing – review & editing:** Thomas M. Herndon, Cristina Ausin, Nina N. Brahme, Sarah J. Schrieber, Michelle Luo, Frances C. Andrada, Carol Kim, Wanjie Sun, Lingjie Zhou, Stella Grosser, Sarah Yim, M. Stacey Ricci.

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
