## [Decision Letter · Decision Letter 0]

6 Sep 2023

PONE-D-23-17848Safety outcomes when switching between biosimilars and reference biologics

 A systematic review and meta-analysisPLOS ONE

Dear Dr. Herndon,

Thank you for submitting your manuscript to PLOS ONE. After careful consideration, we feel that it has merit but does not fully meet PLOS ONE’s publication criteria as it currently stands. Therefore, we invite you to submit a revised version of the manuscript that addresses the points raised during the review process.

We look forward to receiving your revised manuscript.

Kind regards,

Enrique Teran

Academic Editor

PLOS ONE

Journal Requirements:

Additional Editor Comments:

Dear Authors,

Thank you for your submission and we are sorry it has been longer than expected to provide feedback on you manuscript. Following some comments from the reviewers for your consideration before a final decision:

The authors have analysed the impact of original-to-biosimilar switch or vice versa in randomized controlled studies (or extension studies), both from US FDA data bases, and published information. A statistical analytical approach has been applied to assess the impact of single or multiple switches on safety, including immunogenicity. No differences in the immunogenicity and safety profile have been identified in 5,252 patients included in the analysis.

General Comments-suggestions

The authors have put together a rigorous scientific manuscript that is worth being published. Reassuring healthcare professional based on evidence is of outmost relevance in order to foster biosimilar penetration and benefits.

While a few papers (meta-analysis) have been published on RWE safety outcomes as a result of switching from the original reference medicine to a biosimilar, this manuscript is of particular relevance given the authors affiliation, US FDA, the origin of the data analysed - US sourced data of controlled trials- and the statistical approach undertaken.

Specific Comments-suggestions (P = Page / L = Line)

A few comments are made below. Food for thought rather than actual major issues.

P2 L25

“Biosimilars are increasingly available for the treatment of many serious disorders, however concerns persist about switching a patient to a biosimilar whose condition is stable while on the reference biologic” I would suggest to slightly rephrase this sentence, or at the very least, put it into context.

Although certainly concerns have been raised, they are (1) overall progressively decreasing, (2) among healthcare market stakeholders mostly limited to prescribers (some prescribers and, as a results, patients), (3) probably more prevalent in the US than in Europe, or than in other reference jurisdictions with a more extended experience with biosimilars and (4) focused on complex biologics with specific safety profiles. It may sound a bit alarming and cause a to raise such upfront. In particular given the fact that the conclusion is that concern are unfounded based on evidence. (Same P3 / L39 and P20 / L365).

P3 L48

“Increasing the availability and use of biosimilars is an important public health strategy for reducing drug costs and increasing the availability of biological products to underserved populations” The authors may want to consider and state other benefits some using biosimilar may be worth

being mentioned in spite of the fact that the manuscript focuses on patient access (foster innovation, prevent shortages, etc).

Material and methods section

- It is assumed that all the switching studies included in the analysis were conducted in order to claim interchangeability designation. Correct? Did any of the studied biosimilar candidates analysed not obtain interchangeability designation?

- Forty four distinct STPs were identified from thirty-one unique studies. This is an important piece of information since there may be a patient, centre, protocol and/or investigator’s impact on the outcome. Is this accounted for in the statistical analysis? For instance a patient undergoing a switch and claiming an ADR may again claim an ADR in a different STP (possibly owing to nocebo effect).

Results and discussion

- No differences have been found, yet nocebo effect is found in clinical practice Nosuch an effect is observed in RCT?

P18 L311

“The findings in this report raise several timely questions regarding the data needed to support approval of biosimilars and how to address regulatory requirements unique to the BPCI Act for the “interchangeable” designation..” How about using this kind of studies in order to withdraw the interchangeability designation in the long run? Would that be appropriate/feasible?

P19 L337

“Limitations of this review include the small number of patients in the safety evaluations from some source material..” Isn’t the number of patients being taken not consideration in the statistical analysis?

Reviewers' comments:

Reviewer's Responses to Questions

**Comments to the Author**

1. Is the manuscript technically sound, and do the data support the conclusions?

Reviewer #1: Yes

2. Has the statistical analysis been performed appropriately and rigorously? 

Reviewer #1: Yes

3. Have the authors made all data underlying the findings in their manuscript fully available?

Reviewer #1: Yes

4. Is the manuscript presented in an intelligible fashion and written in standard English?

Reviewer #1: Yes

5. Review Comments to the Author

Reviewer #1: Please, see file attached

Ms: PONE-D-23-17848

“Safety outcomes when switching between biosimilars and reference biologics. A systematic review and meta-analysis”

The authors have analysed the impact of original-to-biosimilar switch or vice versa in randomized controlled studies (or extension studies), both from US FDA data bases, and published information. A statistical analytical approach has been applied to assess the impact of single or multiple switches on safety, including immunogenicity. No differences in the immunogenicity and safety profile have been identified in 5,252 patients included in the analysis.

General Comments-suggestions

The authors have put together a rigorous scientific manuscript that is worth being published. Reassuring healthcare professional based on evidence is of outmost relevance in order to foster biosimilar penetration and benefits.

While a few papers (meta-analysis) have been published on RWE safety outcomes as a result of switching from the original reference medicine to a biosimilar, this manuscript is of particular relevance given the authors affiliation, US FDA, the origin of the data analysed - US sourced data of controlled trials- and the statistical approach undertaken.

6. PLOS authors have the option to publish the peer review history of their article (what does this mean?). If published, this will include your full peer review and any attached files.

Reviewer #1: **Yes: **Fernando de Mora

---

## [Author Response · Author response to Decision Letter 0]

13 Sep 2023

COMMENT 1

P2 L25 of original submission

Biosimilars are increasingly available for the treatment of many serious disorders, however concerns persist about switching a patient to a biosimilar whose condition is stable while on the reference biologic” I would suggest to slightly rephrase this sentence, or at the very least, put it into context.

Although certainly concerns have been raised, they are (1) overall progressively decreasing, (2) among healthcare market stakeholders mostly limited to prescribers (some prescribers and, as a results, patients), (3) probably more prevalent in the US than in Europe, or than in other reference jurisdictions with a more extended experience with biosimilars and (4) focused on complex biologics with specific safety profiles. It may sound a bit alarming and cause a to raise such upfront. In particular given the fact that the conclusion is that concern are unfounded based on evidence. (Same P3 / L39 and P20 / L365). See other edits at end of current page 3.

RESPONSE TO COMMENT 1

We agree with the suggestion. In response, we have made a small addition to the Abstract (P2 L24 of Tracked Changes version) and added a sentence to the Introduction (P3 L60 of Tracked Changes version).

COMMENT 2

P3 L48 of original submission

Increasing the availability and use of biosimilars is an important public health strategy for reducing drug costs and increasing the availability of biological products to underserved populations” The authors may want to consider and state other benefits some using biosimilar may be worth being mentioned in spite of the fact that the manuscript focuses on patient access (foster innovation, prevent shortages, etc).

RESPONSE TO COMMENT 2

We agree with the statement that there are many potential benefits associated with the increased availability of biosimilars, but we do not consider this the focus of the work contained in the submission. Attribution of additional benefits beyond the two we have cited is a complex topic and we are hesitant to make statements that may be misinterpreted. We respectfully decline to make additional edits in response to this comment.

COMMENT 3

Material and methods section

It is assumed that all the switching studies included in the analysis were conducted in order to claim interchangeability designation. Correct? Did any of the studied biosimilar candidates analysed not obtain interchangeability designation?

RESPONSE TO COMMENT 3

The Switch Treatment Periods (STPs) included in the analyses were taken from studies designed to achieve a variety of scientific and/or regulatory objectives. Seeking the Interchangeable designation was among these. We are unable to comment on the goals of specific programs.

COMMENT 4

Forty four distinct STPs were identified from thirty-one unique studies. This is an important piece of information since there may be a patient, centre, protocol and/or investigator’s impact on the outcome. Is this accounted for in the statistical analysis? For instance a patient undergoing a switch and claiming an ADR may again claim an ADR in a different STP (possibly owing to nocebo effect).

RESPONSE TO COMMENT 4

Individual data are generally not available and what we have are summary statistics. Therefore, we could not conduct an ideal individual-level statistical analysis that accounts for the intra-subject correlation between different STPs within the same individual. However, we did conduct multiple Logistic Regressions to evaluate the impact of summary demographic/study design factors on the odds ratio (OR) of safety events between Switch and No Switch, e.g., number of STPs in a study, number of switches during STP, study design, patient population, duration of Period 1, duration on study drug before switch, mean age, gender %, and mean BMI. It turns out that none of these changed our conclusion: adjusted ORs of safety events between Switch and No Switch are all similar to the unadjusted OR.

COMMENT 5

No differences have been found, yet nocebo effect is found in clinical practice No such an effect is observed in RCT?

RESPONSE TO COMMENT 5

We agree with the suggestion that the nocebo effect, a patient’s perceived attribution of an undesirable side effect to an intervention for which they have a negative expectation is observed in RCT as well as RWS and clinical practice. Our opinion is that blinding in a RCT will equalize the nocebo effect across study arms.

COMMENT 6

P18 L311 of original submission

The findings in this report raise several timely questions regarding the data needed to support approval of biosimilars and how to address regulatory requirements unique to the BPCI Act for the “interchangeable” designation.” How about using this kind of studies in order to withdraw the interchangeability designation in the long run? Would that be appropriate/feasible?

RESPONSE TO COMMENT 6

We thank the reviewer for the comment. Changing the interchangeable designation is a legislative matter and outside the scope of this work.

COMMENT 7

P19 L337 of original submission

Limitations of this review include the small number of patients in the safety evaluations from some source material.” Isn’t the number of patients being taken not consideration in the statistical analysis?

RESPONSE TO COMMENT 7

We appreciate the reviewer raising this point. We have added additional text to the Discussion section (P19 L339 of Tracked Changes version) to address the comment.

---

## [Editor Report · Decision Letter 1]

18 Sep 2023

Safety outcomes when switching between biosimilars and reference biologics

 A systematic review and meta-analysis

PONE-D-23-17848R1

Dear Dr. Herndon,

We’re pleased to inform you that your manuscript has been judged scientifically suitable for publication and will be formally accepted for publication once it meets all outstanding technical requirements.

Kind regards,

Enrique Teran

Academic Editor

PLOS ONE

---

## [Editor Report · Acceptance letter]

25 Sep 2023

PONE-D-23-17848R1 

Safety outcomes when switching between biosimilars and reference biologics
 A systematic review and meta-analysis 

Dear Dr. Herndon:

I'm pleased to inform you that your manuscript has been deemed suitable for publication in PLOS ONE. Congratulations! Your manuscript is now with our production department. 

Kind regards, 

on behalf of

Dr. Enrique Teran 

Academic Editor

PLOS ONE